# Smooth-edged Perturbations Improve Perturbation-based Image Explanations

Anonymous Full Paper
Submission 24

## Abstract

Perturbation-based post-hoc image explanation methods are commonly used to explain image prediction models by perturbing parts of the input to measure how those parts affect the output. Due to the intractability of perturbing each pixel individually, images are typically attributed to larger segments. The Randomized Input Sampling for Explanations (RISE) method solved this issue by using smooth perturbation masks.

While this method has proven effective and popular, it has not been investigated which parts of the method are responsible for its success. This work tests many combinations of mask sampling, segmentation techniques, smoothing, and attribution calculation. The results show that the RISE-style pixel attribution is beneficial to all evaluated methods. Furthermore, it is shown that attribution calculation is the least impactful parameter. The implementation of and data gathered in this work is available online [1].

## 1 Introduction

Over the past decade, deep neural networks (DNN) have proven proficient at solving computer vision tasks [1]. However, the black-box nature of DNNs causes issues, including difficulties in understanding when the model is wrong, lack of trust in the models, and legal issues [2]. The goal of the field of Explainable Artificial Intelligence (XAI) is to make AI models more transparent to mitigate these issues.

Some research in XAI focuses on developing models that are inherently explainable [3]. Other research uses so-called global methods that attempt to explain the entirety of a model's prediction space [4]. However, these approaches are not suitable for DNNs. A popular approach that avoids these problems is post-hoc explanations [5].

Post-hoc explanations forego trying to understand the model in its entirety and focus instead on explaining individual predictions. For example, instead of explaining the entire process by which a bank makes loan decisions the banker only needs to explain the parts of the process that are important for a given decision. One type of post-hoc explanation that is popular in the computer

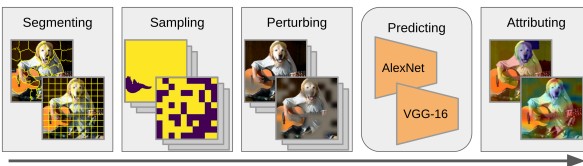

**Figure 1.** The pipeline for perturbation-based image attribution used in this work. The image is segmented, samples indicating what segments to perturb are drawn, the sampled segments are perturbed, the model to explain makes predictions for the perturbed samples, and the input-output pairs are used to calculate attribution per-segment and per-pixel.

vision domain is perturbation-based explanations. Perturbation-based explanations work by analyzing how the model's predictions change as the original input is perturbed. As they only need the given inputs and outputs perturbation-based explanations are model-agnostic and can be applied to any model.

Since the information in images is generally found in the relationships between many pixels [6], perturbing individual pixels is unlikely to have much impact on the prediction. As such, perturbations are typically made on larger pixel segments. Depending on the method these segments are either perturbed one at a time or several at once with different sampling methods for determining what segments to perturb.

The general pipeline for calculating perturbation-based image explanations consists of segmenting, sampling, perturbing, predicting, and attributing, as shown in Fig. 1. The image is split into segments and a number of samples are drawn indicating which segments should be perturbed. For each sample, a new image is created by perturbing the indicated segments in some way. Perturbation often consists of occluding the segments with a solid color [7], but other distortions such as inpainting have also been used [8]. The model output from these perturbed inputs can then be used to attribute influence to the segments based on how the output changes when they are perturbed or not. There are many ways to calculate attribution based on the input-output pairs, such as average output when a segment is included [9] or excluded [10]. Another method is to train a surrogate model to predict the output based on the perturbations and use the learned parameters as attribution [11, 12].

Since attribution is calculated based on which

---
[1]Removed for anonymization

**Table 1.** The different parameters of the perturbation-based image explanation pipeline used in this work.

| Segmenting + Perturbing | Sampling | Samples | Model | Attribution |
|---|---|---|---|---|
| Grid+Bilinear | Random | 4000/8000 | AlexNet | CIU |
| Grid+Gaussian | Entropic | 400 | VGG-16 | PDA |
| SLIC+Gaussian | Only one | 50 | ResNet | LIME |
| | All but one | | | SHAP |
| | | | | RISE |

segments are perturbed, most methods assign attribution per-segment, but cannot differentiate the influence between pixels. The Randomized Input Sampling for Explanations (RISE) method solves this by using smooth perturbations, where pixels are perturbed more as they get closer to the segment center [9]. This is then used to calculate a per-pixel attribution by weighing the attribution of a pixel by how perturbed the pixel was.

Like many perturbation-based explanations, RISE is introduced as an entire pipeline from segmenting to attribution. This work explores how the benefits of smooth-edged perturbations can benefit other perturbation-based pipelines. It also expands on the original RISE implementation by evaluating a variety of segmentation, sampling, perturbation, and attribution methods using occlusion metrics [8]. The evaluations are carried out on the ImageNet validation set [13] for three different CNNs [14–16] using both per-segment and per-pixel attributions. The different pipeline parameters that have been combined and evaluated are shown in Table 1.

The results show that using smooth edges and weighing pixels by how faded they are in a given sample improves the performance of all evaluated methods. Another noteworthy result is that the method of calculating the attribution, which is typically what is highlighted as the most important part, has little impact on performance. Conversely, the sampling, number of samples, segmentation, and per-pixel attribution all have a greater impact on performance.

## 2 Methodology

This work evaluates pipelines using all possible combinations of the different segmenting, sampling, perturbing, and attribution methods as well as sample sizes listed in Table 1. Each pipeline is tested with three different ImageNet [13] pretrained CNNs by using them to explain the models' predictions on the ImageNet validation set and then evaluating those explanations using occlusion metrics [8]. The different parts of the experiments are described in detail in the following subsections.

### 2.1 Segmenting

This work evaluates two segmenting techniques; grid and SLIC [17] segmentation. Grid segmentation splits the image a given number of times horizontally and vertically. SLIC is a rule-based algorithm that iteratively calculates segment "centers", assigns each pixel to the closest center in a color-position space, and recalculates the segment centers repeatedly until convergence.

The experiments use the same $7 \times 7$ grid of segmentation as the original RISE implementation [9]. To make the SLIC segmentation as similar to the grid implementation as possible, SLIC is instantiated with 49 segment centers in the experiments. The default scikit-image implementation for SLIC is used [18].

### 2.2 Sampling

This work generates samples indicating which segments to perturb using random, entropic, "only one", and "all but one" sampling. Random sampling consists of, for each segment and sample, randomly deciding whether it should be perturbed with a probability $p$. In this work $p = 0.5$.

Entropic sampling is created to be similar to the default KernelSHAP sampling behavior [12]. Entropic sampling will first sample the low-entropy samples, i.e. samples with as many or as few segments perturbed as possible. No segments are perturbed in the first sample, all segments are perturbed in the second, followed by all possible combinations of one segment perturbing and all combinations of one segment unperturbed, followed by combinations of two segments perturbed/unperturbed, and so on.

"Only one" and "All but one" sampling consists of creating all samples where only one segment is perturbed and where all but one segment is perturbed respectively. Both methods also add the sample where no segments are perturbed as this is needed by the Contextual Importance and Utility (CIU) attribution [19].

Random and entropic sampling are evaluated for three different sample sizes. The 4000/8000 sample size is used to be consistent with the original RISE evaluation. AlexNet and VGG-16 use 4000 samples and ResNet models were evaluated with 8000 samples. A sample size of 50 is used with all four methods, where "only one" and "all but one" sampling will always create one more sample than the number of segments.

### 2.3 Perturbing

Perturbing consists of pixel-wise multiplication between the normalized image and a perturbation mask of values between 0 and 1. The mask is created by setting all values in the segments to be perturbed to

0 and all others to 1, then the mask is smoothened so that the values closer to the center of each segment are close to 0 and those at the edges and beyond are closer to 1. Thus pixels outside the perturbed segments are mostly unchanged, but fade towards the normalization mean as they get closer to the segment centers. The original implementation achieves this by using bilinear upsampling to scale a $7 \times 7$ grid of 0s and 1s to the size of the full image, an implementation that is replicated in this work. An issue with this method is that it relies on having a lower resolution mask to upscale which excludes using some popular segmentation methods such as SLIC. To combat this issue another method of creating smooth segment masks through applying a Gaussian filter is introduced. For this work, the Gaussian filter has a $\sigma = 10$ which gives similar masks when compared to bilinear upscaling.

## 2.4 Attributing

This work evaluates five existing attribution methods, CIU [19], PDA [10], LIME [11], SHAP [12], and RISE [9]. Some of these methods cover more parts of the pipeline than just attribution. However, in this work, the method names are used as a shorthand for the attribution calculation from the input-output pairs created by the predicting step of the pipeline.

CIU is one of the oldest XAI methods [19] with more recent works implementing it for images [20]. CIU works by calculating the Contextual Importance ($CI$) of a feature $s$ as $CI_1(s) = \frac{max(Y, Y\setminus_s) - min(Y, Y\setminus_s)}{max(Y\setminus) - min(Y\setminus)}$, where $Y$ is the original output, $Y\setminus_s$ is all the outputs when feature $s$ has been perturbed, and $Y\setminus$ are all outputs. The CIU implementation for images [20] instead calculates the importance of a segment $s$ by perturbing all other segments ("all but one" sampling). In this work this is calculated as $CI_2(s) = \frac{max(Y, 1-Y\setminus_{\bar{s}}) - min(Y, 1-Y\setminus_{\bar{s}})}{max(Y\setminus) - min(Y\setminus)}$, where $Y\setminus_{\bar{s}}$ is all the outputs where $s$ is not perturbed. The Contextual Utility ($CU$) of the feature $s$ is then calculated as $CU(s) = \frac{Y - min(Y\setminus_s)}{max(Y\setminus) - min(Y\setminus)}$ where $Y$ is the original output. The attribution score for the feature $s$ is calculated in this work as $w_{CIU}(s) = CI(s) \cdot (CU(s) - 0.5)$. While there are implementations of CIU that handle change in more than one feature at a time [21], they are not compatible with the evaluation used in this work. As such, CIU is only evaluated for the "only one" and "all but one" sampling methods using $CI_1$ and $CI_2$ respectively.

Prediction Difference Analysis (PDA) [10] works similarly to CIU, but uses average difference instead of maximum difference. PDA has been adapted to work with images [10], though both in the original and image implementation only a single feature is changed at a time. In this work, PDA has been generalized to work when multiple features are perturbed

simultaneously. The PDA attribution is given by $w_{PDA}(s) = Y - avg(Y\setminus_s)$.

Locally-Interpretable Model-agnostic Explanations (LIME) [11] was originally introduced as an umbrella term used to cover any instance where a single prediction is explained by training an interpretable model to mimic the original model's prediction in the neighborhood of the original input. However, LIME has since been associated with specifically training a linear surrogate model [3, 7] as this is how the method was demonstrated originally. In this work, LIME is implemented as a linear model $y = b + \sum_{s \in S} w_s \cdot x_s$, where $y$ is the output of the model, $b$ and $w_s$ are the learned bias and weights, and $x_s = 0$ if the segment is perturbed and 1 otherwise. The attribution of LIME for segment $s$ is the value of $w_s$ after the linear model has been fit to the input-output pairs using least squares.

Kernel SHAP [12] is a modification to LIME such that, under certain assumptions, the weights learned by the linear model will tend towards the Shapley values [22] scoring how the features contribute to the prediction. This is achieved by scaling the input-output pairs with a kernel function $\pi(X) = \frac{|S|-1}{\binom{|S|}{|X|}|X|(|S|-|X|)}$, where $|s|$ is the number of segments and $|X| = \sum_{x_s \in X} x_s$. As such the SHAP values can be retrieved by solving $\pi(X)y = b + \sum_{s \in S} w_s \cdot \pi(X)x_s$ using least squares.

The attribution used by RISE [9] is similar to PDA, but instead of using the average decrease when the feature is perturbed, it uses the average prediction when it is not perturbed. RISE attribution for a segment is given by $w_{RISE}(s) = avg(Y\setminus_{\bar{s}})$.

Additionally, RISE attribution utilizes smooth pixel perturbation masks to assign per-pixel attributions according to $w_p = \frac{1}{\sum_{s \in S} M_s^p} \sum_{s \in S} w_s \cdot M_s^p$, where $M_s^p$ is the value of pixel $p$ in the perturbation mask of segment $s$. Note that this calculation means that pixels outside segment $s$ which were slightly perturbed due to the smooth mask, also include that influence in the calculation. For example, this means that pixels at segment borders get a lesser influence from both segments. This work evaluates attribution both per-segment ($w_s$) and per-pixel ($w_p$).

## 2.5 Evaluation

The various pipelines are tested by explaining the predictions of three ImageNet pretrained CNNs on the ImageNet validation set and evaluating those explanations with occlusion metrics. The three pretrained CNNs are AlexNet [14, 23], VGG-16 [15], and ResNet-50 [16] using trained parameters from the Torchvisio 0.15.2 framework [24]. The input to the models is normalized using the average pixel values of ImageNet.

Evaluation is carried out using one image per class of the ImageNet validation for a total of 1000 images

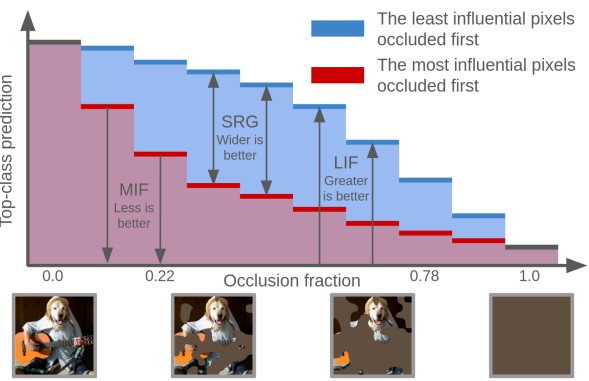

**Figure 2.** Showcase of how LIF, MIF, and SRG metrics are calculated by steadily occluding the least or most influential pixels of an image and calculating the value of the top class predicted for the original image.

(2% of the total validation set). Limited evaluation was performed on the full validation set and no statistically significant ($p < 0.05$) difference could be found compared to using 2% of the data. For each image, the top predicted class of each model was explained through segment and pixel attribution using each pipeline. The attribution was then evaluated using occlusion metrics. The occlusion metrics used in this work are similar to the ones used for evaluating the original RISE implementation [9] though modified to take advantage of recent findings that increase the consistency [8].

Occlusion metrics consist of increasingly occluding the image and observing how the prediction changes. By occluding the pixels with the Least Influence First (LIF), the model prediction is expected to be similar until the influential pixels start getting occluded. Conversely, by occluding the pixels with the Most Influence First (MIF), the model prediction is expected to lower quickly. A good explanation should have a large area under the LIF prediction-occlusion curve and a small area under the MIF curve. LIF and MIF are equivalent to the insertion and deletion metrics used to evaluate RISE originally. The LIF and MIF metrics are highly variable but the combined metric ($LIF - MIF$), called Symmetric Relevance Gain (SRG), is more reliable [8]. The connection between the three metrics is visualized in Fig. 2.

This work uses the SRG metric to evaluate performance. It is calculated by occluding the image over a total of 10 equal steps (from 0% occlusion in step 1 to 100% occlusion in step 10). The remaining pixels with the lowest or greatest attribution score for original top-class prediction are occluded in each step for LIF and MIF respectively. When there are many pixels with the same attribution, then pixels are chosen in an arbitrary deterministic order. Occlusion is performed by setting the pixels to the mean pixel value of the image, which mirrors one of the evaluation methods explored by Blücher et

**Table 2.** The average SRG in % for all pipelines with different combinations of segmenting, perturbing, and attribution methods with either per-segment or per-pixel attribution.

| Segmenting + Perturbing | Attribute per | CIU* | PDA | LIME | SHAP | RISE |
|---|---|---|---|---|---|---|
| Grid+bilinear | Segment | 11.7 | 14.9 | 14.9 | 14.5 | 15.9 |
| Grid+bilinear | Pixel | 14.1 | 16.3 | 16.5 | 16.4 | 17.6 |
| Grid+Gaussian | Segment | 11.6 | 14.9 | 15.0 | 14.6 | 15.8 |
| Grid+Gaussian | Pixel | 14.4 | 16.5 | 16.8 | 16.7 | 17.8 |
| SLIC+Gaussian | Segment | 15.7 | 17.1 | 17.4 | 17.5 | 18.0 |
| SLIC+Gaussian | Pixel | 16.8 | 17.6 | 18.2 | 18.3 | 18.8 |

*CIU is not evaluated for random or entropic sampling, which have greater average performance.

al. [8]. The average of the original top-class prediction over these 10 images is then recorded as the LIF and MIF scores. The SRG score is calculated as $LIF - MIF$.

## 3 Results and Analysis

The results consist of the LIF, MIF, and SRG metrics for every attribution pipeline. As this is too much data to present in this work, it is summarized as the average SRG metric for different parameter combinations. The complete data is available in spreadsheet form, where tables like those below can easily be generated[2].

The results of different combinations of segmenting, perturbing, and attribution as the average SRG metric can be found in Table 2. Notably, for all combinations of segmenting, perturbing, and attribution methods using per-pixel instead of per-segment attribution improves performance. Furthermore, the improvement of using per-pixel rather than per-segment is significantly greater than switching attribution methods. Using a Gaussian filter instead of bilinear upsampling does not affect performance, except for a mild increase in SRG. SLIC performs much better than Grid segmenting in all cases but sees a relatively smaller improvement when using per-pixel attribution. This is likely due to SLIC having better boundaries between segments.

The average SRG for pipelines with different sampling methods and sample sizes over the different attribution methods is shown in Table 3. Unsurprisingly, increasing sample size yields improved performance. What is surprising is that random sampling significantly outperforms entropic and does so even for SHAP for which it is specifically adapted. PDA struggles with entropic sampling, except for when the sample size is 50, which is almost equivalent to "only one" sampling. Again it is noteworthy that the attribution method is the least impactful factor, except under some combinations of sampling

---

[2]Removed for anonymization

**Table 3.** The average SRG in % for all pipelines with different combinations of sampling and attribution methods.

| Sampling | Sample size | CIU | PDA | LIME | SHAP | RISE |
|----------|-------------|-----|-----|------|------|------|
| Random | 4000/8000 | - | 25.6 | 25.8 | 24.0 | 25.6 |
| Entropic | 4000/8000 | - | 14.7 | 18.2 | 18.8 | 17.8 |
| Random | 400 | - | 22.9 | 24.1 | 22.3 | 22.9 |
| Entropic | 400 | - | 9.0 | 15.6 | 17.3 | 15.0 |
| Random | 50 | - | 16.0 | 6.8 | 6.8 | 16.0 |
| Entropic | 50 | - | 13.3 | 13.3 | 13.3 | 13.3 |
| Only one | 50 | 13.3 | 13.3 | 13.3 | 13.3 | 13.3 |
| All but one | 50 | 14.9 | 14.9 | 14.9 | 14.9 | 14.9 |

and sample size that seem to cause some attribution methods to fail. All attribution have the same performance for Entropic, "only one" and "all but one" sampling with a sample size of 50 as under these limited circumstances the order of influential segments is equivalent for CIU, PDA, and RISE. At the same time, LIME and SHAP converge to the same ordering.

## 4 Discussion

This work shows that the smooth-edged masks used in the original RISE implementation can be modified to work with many different attribution pipelines and that this improves performance on occlusion metrics. However, occlusion metrics do not necessarily correlate with usefulness to humans. It may be the case that per-pixel attribution simply gives advantages in performance calculation that are not noticeable in user testing, for which further work is needed.

The results also show that each part of the pipeline that is explored can have a significant impact on performance. Most works that introduce some form of perturbation-based image explanations often introduce an entire pipeline but do not examine the parameters of that pipeline separately. This leads to a poor understanding of what makes one method better, especially when later works compare those pipelines against each other [25, 26]. Contrastingly, this work along with Blücher et al. [8] shows how the different parameters can be analyzed independently.

The evaluation in this work relied on the explanation methods being separable into different parameters that could be combined in various ways. This is not always the case, even if the method otherwise produces sound explanations. For example, the original RISE shifts the perturbation masks by some pixels so as not to center the same pixels every time. This approach works with the RISE attribution method since it can directly assign influence to pixels. However, this is not feasible for other methods, as such shifting could not be evaluated with

the experiments conducted in this work. Additionally, the use of occlusion metrics requires attribution scores for individual features. For example, the CIU method can be used when combinations of features are perturbed simultaneously, however, those explanations instead give attribution to how beneficial the combinations are, rather than splitting the influence between the features. Another example is using decision trees as surrogate models. Decision trees are typically interpretable but do not assign influence to features directly.

A general issue with all current perturbation-based methods is that they require that the model be run multiple times. This inevitably scales the computation needed by at least a factor equal to the sample size used. With ever-increasing computational demands by newer DNN models, even a low sample size let alone thousands of samples, may be unrealistic to presume for an explanation of a single decision. Developing perturbation-based methods that can give good explanations with low sample size is therefore a promising future direction. In some cases, such as medical diagnosis prediction, the need for and the value of explanations are likely high enough that it is worth increasing computational demands by factors of thousands.

One contender to perturbation-based methods is gradient-based methods. Gradient-based post-hoc methods utilize that DNNs are typically differentiable and use the gradients of the prediction to calculate an explanation. This gives gradient-based methods the advantage that they often do not need multiple calls to the model. However, gradient-based explanation methods, especially in the computer vision domain, have multiple times been shown to be unreliable [27–29]. Perhaps a combination of the different post-hoc paradigms could benefit from the reliability of perturbation-based methods and the lower computational demands of gradient-based methods. For example, the initial gradient-based explanation could inform the optimal segments or samples to use with a permutation-based approach.

Ultimately, the true measure of any explanation is its usefulness to humans. For example, a prior study found that users preferred CIU explanations to LIME and SHAP [20], which is not obvious from the results in this work. However, the number of different parameter combinations that exist in XAI is too many for human evaluators. As such, future works might strive to use metrics such as SRG to find the best candidate pipelines and then compare those using human evaluation. Such experiments would require an additional step to the pipeline; communicating. How an explanation is communicated to humans can vary between implementations and is another factor that can disrupt experiments. As such a study focusing solely on communication of image attribution would be beneficial.

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
