# OpenReview forum: "Smooth-edged Perturbations Improve Perturbation-based Image Explanations"
_NLDL.org/2025/Conference — Submitted to NLDL 2025_

### Official Review · Reviewer_WvMh · 2024-09-26
**Smooth-edged Perturbations Improve Perturbation-based Image Explanations**

**Confidence:** 4

**Summary:**

Based on the success of RISE (Randomized Input Sampling for Explanations) in the field of XAI, this paper studies a range of combinations of perspectives, including sampling, segmentation, and XAI methods which the authors call the attribution calculations.
It shows that the RISE-based pixel-based methods are effective, in particular when the sample size is large.
Although it focuses on some specific image datasets and CNN models, this paper reveals the dependency of SRG values on the sampling methods (random/entropic) regardless of the attribution.

**Strengths:**

In this paper, the authors compare a variety of aforementioned combinations. The perspectives in their experiments are clearly described and the results are exhibited. Not only being interesting, it is important to examine how XAI methods work well when the possible combinations of methods and attributions are changed.

**Weaknesses:**

Although this study works on an important topic, it seems that the conclusion (discussion) section is not so clear in the sense that
what has been done when compared to the original purpose is not clear.

For example, in Table 3, it seems that LIME exhibits the best SRG value with 4000/8000 random sampling with the value of 25.8.
However, it's not clear whether this is sufficient or not when we recall the assertion of the abstract stating "the RISE-style pixel attribution is beneficial", because we don't know how we can compare this result to the performance of original LIME.

The authors explain the definition of the SRG value right before Section 3, so I think it is better if the sufficient value for the explanation
is defined aroud there. Moreover, the word "RISE-style pixel attibution" sounds unclear. I think it would be better if this part is re-phrased
in a more general and easy-to-understand one.

A question to the authors: could you elaborate on the relationship between Tables 2 and 3, for I cannot find the method of sampling in Table 2 for example.

**Final Rebuttal Confidence:**

3

**Final Rebuttal Justification:**

I found that some of my concerns may be removed. For example, they say that they have already conducted the hypothesis test to quantitatively support their assertions and that they will make the assertions clearer and check English writing. These would lead to the additional strengths of this paper. However, some of my concerns still exist. (i) They didn't clarify the experimental conditions of Table 2 (sampling method) that was raised in my comment. (ii) They say that increasing the datasets is difficult. Although they say that the evaluation with 2% of the dataset was not significant (p<0.05) with the experiments with limited data, I think it is better then to describe the details on this part and to deduce some statistical indicators such as confidence interval, which would be beneficial to support their assertion. Based on these, I think I'd be better off retaining my original rating (2).

**Justification:**

Although the authors conducted the experiments on the comparison of extensive combinations, my major concern is that the assertion of the paper is not clear. They assert that "the RISE-style pixel attribution is beneficial to all evaluated methods" in Abstract,  but it is not clear from the results, mainly Tables 2 and 3, for they did not compare the original performance and the one with the proposed method.
From Tables 2 and 3, it may give an impression that the original RISE is almost sufficient.

Therefore, in order to improve this paper, it seems better to compare the original methods and those proposed in this paper.
Second, the authors should explain why SRG value is good as an evaluation major of XAI methods.

Third, it would be better if the authors employ statistical hypothesis testing when they conclude something; for example, in line 349, they say "improves performance", but it sounds qualitative without talking about the statistical significance. The same applies to lines 363--364.

The results exhibited here are limited only to image datasets, so in Section 4, it would be better to talk about extending this approach to
other types of datasets. In machine learning, the no-free-lunch theorem tells us that no specific model is always effective.
It might be the case to the field of XAI, which implies the possibility of different results depending on the types of the datasets.

The conditions of Tables 2 and 3 should be clarified. For example, in Table 2, I could not find the method of sampling.

Finally, if possible, the authors should polish up the English description in the main text. For instance, "implementation of and data"
in Abstract would be "implementation of data", and "is available online" should be "are available online". There are similar parts
in the main text.

To sum up, I think this paper falls into the region that is below the border of acceptance.

---

> ### Author Rebuttal · Authors · 2024-10-24
>
> Thanks for the honest review. Below you can find our responses to the issues that you have brought up followed by what we will do to address them.
>
> ## General clarity of purpose
> After reading your review and the other reviews we have realized that a key issue with our manuscript is lacking clarity regarding what is being investigated and what has been found. The core of our work is an ablation study of the different parts of the perturbation-based pipeline used by RISE (but also many other methods, though this work uses the RISE pipeline as a base). From this we get results showing which parts of the pipeline most affect performance in terms of SRG insertion/deletion metric. One part of the pipeline that has a large impact is the pixel attribution called "pixel" in Table 2. We realize that we mix up the investigation and the finding (making it seem as though we are only investigating pixel attribution which is only a part of the pipeline being investigated.
>
> To address this we will rewrite the manuscript to better describe the ablation study that is actually being performed and separate the study (which part of the pipeline is important for performance) from its findings (that pixel attribution is good). This will also be reflected in the discussion section which will refer back to the better description of the ablation study and list what is impactful, what this means, and how this impacts the field.
>
> ## Comparison with original methods
> How the results compare to the original methods is difficult to tackle as this is an ablation study of the pipeline used by these methods for images. LIME and SHAP are only attribution methods that do not inherently have the other parts of the pipeline which instead have to be decided for when implementing the for images. As such any of the many pipelines examined that use LIME or SHAP is true to the original work. CIU and PDA are methods that define specific sampling and attribution methods, leaving the rest to depend on implementation. As such any of the pipelines that use CIU with "only one" or "all but one" sampling and PDA with "only one" sampling is true to the original work. Finally, RISE defines the entire pipeline that is being investigated. While we do not compare to the exact original RISE implementation (which could not be put into the pipeline used for the ablation study) we do have one which is analogous to the original work (Grid+bilinear, random, 4000/8000, any model, RISE). To use your example, this means that comparing the original LIME with LIME using pixel attribution is as easy as checking the LIME column of Table 2 and comparing the attribute per segment (original) to the attribute per pixel performance.
>
> To address this we will:
> * Better describe the ablation study and how the original versions of the attribution methods used relate to the different parts of the pipelines.
> * Better describe the Tables and what the results found in them refer to.
> * Add the performance of the pipelines corresponding to the original implementations of these methods.
> * If desired, run experiments with the exact original RISE implementation for comparison.
> * Rephrase "RISE style pixel attribution" to simply "per pixel attribution" to use consistent language between the text and tables.
>
> ## Justify SRG
> We use SRG as it is an insertion/deletion metric which is one of the most common image attribution evaluation methods. The original work uses LIF/MIF (though under other names). However, a recent work has shown that LIF/MIF by themselves are not consistent between different occlusion types, but by combining the two into SRG a metric that gives similar rankings of methods regardless of occlusion method is achieved.
>
> We will address this by clearly stating the purpose for using SRG in the manuscript.
>
> ## Limited datasets
> This work is explicitly about image attribution so moving beyond image datasets would not be possible, however, more image datasets with different domains or tasks is an option. We use the dataset on which the original RISE implementation was evaluated. More datasets would be useful for drawing broader conclusions. Unfortunately, these experiments are very time-consuming to perform and as such we had to choose between testing more features or using more datasets. There is not enough time for us to include an evaluation using another dataset.
>
> We will address this by more clearly motivating the use of the ImageNet dataset.
>
> ## Lack of statistical significance testing
> We have run significance testing and the results are significant at p<0.05. This will be added to the tables by bolding the results which are statistically significantly better (unless you have another suggestion for how to integrate this in the results).
>
> ## English grammar and spelling
> We will make sure to be more thorough in spellchecking proofreading in the revised version of the manuscript.

---

### Official Review · Reviewer_VRaE · 2024-10-08
**Concerns on evaluation robustness and connection to human interpretability**

**Confidence:** 3

**Summary:**

The submitted paper explores perturbation-based post-hoc image explanation methods used in deep learning, focusing on explaining model predictions by occluding parts of an input image and measuring the impact on the model's output. The paper builds on existing techniques such as Randomized Input Sampling for Explanations (RISE), by evaluating the effectiveness of smooth perturbation masks across different explanation pipelines. Its key contribution is an evaluation of how smooth perturbation masks perform across different frameworks, testing combinations of segmentation, perturbation, sampling, attribution, and models. The authors analyze how each element in the pipeline influences explanation quality, using Symmetric Relevance Gain (SRG) as the primary evaluation metric.

**Strengths:**

1. The authors perform a detailed evaluation of various parameters involved in the perturbation-based explanation pipeline. By testing combinations of segmentation techniques, sampling methods, and attribution calculation approaches, the paper offers a thorough empirical analysis. This level of experimentation and comparison provides insights into which parts of the pipeline contribute most to performance, contributing to a deeper understanding of these techniques.
2. The methods are evaluated on multiple well-established CNN architectures (AlexNet, VGG-16, ResNet-50), which ensures that the proposed techniques are broadly applicable across different model types.
3. Code and data is available, enabling reproducibility.

**Weaknesses:**

I have concerns regarding the evaluation, particularly with the use of the term "interpretability." The evaluation appears to focus more on the model's sensitivity to input perturbations rather than providing a true measure of interpretability. To support claims of interpretability, it would be valuable to assess whether the explanations generated by the model are understandable and align with human intuition, potentially by involving domain experts in the evaluation process.

Given the cost of involving humans, an alternative could be to conduct experiments using synthetic data in controlled environments. With synthetic data, the ground truth of which features should influence a prediction is known, allowing for a more rigorous comparison of the method's explanations against these known features.

Here are my main questions and concerns:

1. The act of occluding parts of an image introduces perturbations that could affect the model in ways that aren’t directly related to how it normally processes the image, potentially leading to misleading assessments of importance. The way pixels or regions are occluded may introduce unintended biases, making it harder to assess the true impact of the occluded regions on the prediction. It would be beneficial for the paper to provide further elaboration on how these biases are mitigated or accounted for.
2. Based on the methods described, it appears that the occlusion metric primarily measures the sensitivity of the model to the removal of image regions rather than its true explainability. Sensitivity reflects how model outputs change when portions of an image are occluded, but this does not necessarily correspond to human-intuitive explanations. For example, a model might be highly sensitive to patterns in an image's background, which, while influential to the model, might not offer a meaningful explanation to a human observer regarding why a particular prediction (e.g., identifying a "dog") was made. I ask for further clarification on whether the identified regions are inherently meaningful or interpretable for humans. This would strengthen the analysis.
3. How does this relate to causal inference, spurious correlations etc? A model might be overly sensitive to spurious features (e.g., noise or irrelevant patterns), which could lead to explanations that highlight regions that don’t align with meaningful human explanations. How does the approach ensure that it avoids emphasizing these spurious features?
4. The paper sets occluded values to the mean pixel value of the image. However, it would be important to evaluate the sensitivity of the results to other occlusion techniques, such as using the median value, random values, or dataset-wide mean values. Is there a reason for not doing this? For me, it seems that an exploration of how different occlusion strategies affect the explanation would enhance the robustness of the findings.

5. The use of only 1,000 validation images of ImageNet is not sufficiently justified, nor is the decision to use the validation set instead of the test set. The authors should have justified why this sample size is representative and how it was selected.
6.  Evaluating the reliability of the explanation methods is complicated by the lack of ground truth, which would require complete transparency into the model’s decision-making process -- something these explanation techniques are themselves trying to provide. This circular issue should be discussed, and suggestions for addressing it in future work would be appreciated.
7. A comparison with gradient-based post-hoc explanation methods would provide valuable context. Gradient-based methods are well-established in the explainability literature, and a comparison would allow for a more comprehensive evaluation of the proposed method's performance.
8.  It would benefit the clarity of the paper if a formal problem definition were introduced earlier, following the introduction. Additionally, the optimization of different methods could be more broadly contextualized to outline their specific goals and constraints within the broader field of explainability research.

**Justification:**

While the paper makes a meaningful contribution by exploring perturbation-based explanations in detail, its claims of interpretability are weakened by a narrow evaluation framework that primarily measures sensitivity. To strengthen the paper, the authors could explore alternative evaluation metrics:
* insertion and deletion tests to assess the faithfulness of the explanations
* counterfactual explanations to provide causal insights into the model’s decision-making process
* human centric metrics

Further, I ask the authors to justify their experimental choices more clearly.

These additions would allow for a more comprehensive understanding of the proposed methods and their impact on explainability and interpretability.

Moreover, the paper uses only 1,000 validation images from ImageNet without sufficient justification for this sample size. This choice raises concerns about whether the results generalize to a broader dataset, and the decision to use the validation set instead of the test set lacks proper explanation. A more detailed discussion of how this sample size was selected and its representativeness would be beneficial.

---

> ### Author Rebuttal · Authors · 2024-10-24
>
> Thank you for a very thorough review. Below you will find our response to the issues you have identified and how we plan to address them.
>
> ## Clarity of purpose
> After reading your review and the other reviews we have realized that the clarity of what the study is and what its findings are is lacking. The study is an ablation study on the different parts of the perturbation-based image attribution pipeline. One of the key findings of this ablation study is that attribution per pixel (called RISE-style pixel attribution but will be rephrased in the revised version of the manuscript) instead of per segment improves performance as measured by the SRG metric. We are assessing the explanation methods, not the models themselves (which are only there to have something to explain). Furthermore, we are explaining the models, not the data or the ground truth, meaning that what is being communicated to a human is what parts of the image impacted the model's decision. Such attribution maps have been shown to increase human-AI collaborative performance on image classification, so there is a justification for using them for explanation [1]. If the background is influential to the decision this will be shown in the attribution map and a human might then decide to distrust the AI as it does not focus on the salient part of the image. A typical example of this is when a dog or wolf detector was shown to be bad since it focused on the snow in the background rather than the dogs/wolves as it had learned that most images of wolves had snow in them. So showing what part of the image is influential to the model's decision can be useful for informing human users.
>
> We will address this by:
> * Rewriting the manuscript to more clearly make the difference between the ablation study being made and the findings from that study.
> * Integrating the above reference to motivate attribution maps as explanation tools.
> * More clearly state how attribution maps can be used to explain model behavior and that it is the models themselves that are being explained.
>
> ## Measuring interpretability without human evaluation
> True interpretability has to be measured using human evaluation. However, this is a time-consuming task that cannot always be performed. For this reason, many so-called proxy metrics have been devised by the XAI field. For image attribution, the insertion/deletion type metrics are popular and used to evaluate RISE originally. For this reason, SRG, an insertion/deletion metric, was chosen for this work. Additionally, since image attribution claims to find the most influential part of the image, insertion/deletion metrics provide a sanity check as to whether the identified regions are actually the most influential.
>
> Unfortunately, due to the number of experiments being conducted, it is unfeasible to use human evaluation for all the combinations. Instead, perhaps the best-performing combinations could be evaluated in a follow-up study. While a work evaluating insertion/deletion metrics and other XAI metrics would be helpful, it is beyond the scope of this work.
>
> To address this we will explain the proxy metrics and motivate the use of the SRG metric better.
>
> ## Occluding images in abnormal ways and using mean-pixel perturbations
> Yes, using occluding part of the image will create out-of-distribution images which may affect the models in ways not seen in the original dataset. This is somewhat mitigated as all of the explanation pipelines are evaluated in the same way and, on average, are affected by this equally. Additionally, it has been shown that the SRG metric gives similar rankings of explanations regardless of the occlusion used. This means that the more intrusive occlusion used in this work gives a similar performance to using a more natural (but expensive) image generation approach. Occlusion is also performed in the perturbation used by the explanation pipeline. This is a feature/flaw of the method and how to perform this perturbation is a good target for a future ablation study. However, in this work, we use the same perturbation as the original RISE.
>
> We will address this by better explaining the benefits of SRG and suggesting future ablation studies in the description.
>
> ## Causal inference and spurious correlations
> Since image attribution is there to explain why the model made a certain prediction, showing the regions that the model finds influential but are not actually salient to humans is a feature, not a bug. A human seeing an explanation focusing on something irrelevant may use this to understand that the model is likely not trustworthy.
>
> We will address this by more clearly describing how image attribution is used for explanations.
>
> ## Using only 1000 images of the ImageNet validation set
> These experiments are very time-consuming. The 4000/8000 sample size experiments make 8000 model calls per image resulting in 8000000 model calls for one pipeline. In total the experiments required around 4 billion model calls. To save time and energy we decided to use only 1 image per class for the evaluation. The validation set was chosen as this is what was used by the original RISE evaluation. Additionally, it is common in the image explanation field to use even fewer images for evaluation due to the amount of computation resources required by evaluation.
>
> We will address this by describing why we only use 1000 images, how these were selected (the first image of each class in the data set), and why the validation set is being used.
>
> ## Evaluating explanation methods
> As previously stated true XAI evaluation should use human evaluators, but proxy metrics are often used due to the expense of using human evaluators. One category of proxy metrics is to take a model that is inherently interpretable (such as a linear predictor) and see how well the explanation aligns with the model parameters. This type of proxy metric is not readily available to images as a large number of complexly correlated input features make such inherently interpretable models useless for the given task.
>
> We will address this by describing the difference between human metrics and proxy metrics and motivating our use of proxy metrics.
>
> ## Comparison with gradient-based methods
> While comparing to more methods would definitely bring value to the work, this was decided against as they do not fit into the pipeline at all for the ablation study and would bloat the scope of the work. It would be unfair to compare the average performance of some pipelines with the best implementation of a gradient-based method. Instead, this study can be used to inform the creation of better perturbation-based metrics which can be compared against gradient-based metrics in future works.
>
> ## Formal definition of the problem
> As part of our improvement of the clarity of the work we will better describe what the problem is (we do not know which parts of the perturbation-based image attribution pipeline are important for prediction), what we will do about it (an ablation study based around RISE), and what the results are (pixel attribution is an impactful part of the pipeline). We will also make it clearer that while the research is novel and some of the parts being introduced (gaussian blur instead of bilinear) are also novel, in general, we are not introducing a new method, simply making an ablation study of existing ones.
>
> ## References
> [1] Nguyen, G., Kim, D., & Nguyen, A. (2021). The effectiveness of feature attribution methods and its correlation with automatic evaluation scores. Advances in Neural Information Processing Systems, 34, 26422-26436.
>
> [2]Blücher, S., Vielhaben, J., & Strodthoff, N. (2024). Decoupling pixel flipping and occlusion strategy for consistent xai benchmarks. arXiv preprint arXiv:2401.06654.

---

### Official Review · Reviewer_5sjX · 2024-10-09

**Confidence:** 3

**Summary:**

In this paper, the authors provide an analysis on the interaction of different components of the perturbation based feature attribution pipeline.  The authors use existing off-the shelf methods to understand that pipeline which includes segmentation, perturbation and attribution analysis.  Pre-trained CNNs such as VGG and Alexnet are used for evaluation and metrics such as SRG are reported for quantitative evaluation of each combination.

**Strengths:**

The paper provides a detailed overview of perturbation based methods for XAI and illustrates every component well.  The paper has very limited typos.

**Weaknesses:**

1.  The main novelty and key intuitions behind this work is a bit unclear to me.  Is the goal of the paper understanding RISE or is to conduct a benchmark study on the different components of the perturbation based attribution pipeline? Upon reading the paper a few times, I think it is the latter although it is not very clear.

2. How does this approach compare with existing gradient based methods such as Grad-CAM? Although the authors have briefly mentioned this in the discussion section, this is still needed to understand the impact of the contribution.

3. In addition to the SRG metrics, can the authors also utilize other popular metrics such as log-odds ratio[1] to evaluate performance?

4. Can the authors qualitatively contrast between the proposed approach and the related papers [1, 2]?

5. Can the authors explain why the quantitative evaluation has been conducted only on 2% of the validation dataset?


[1]   Schwab, P.; and Karlen, W. 2019. CXPlain: Causal explanations for model interpretation under uncertainty. In Advances in Neural Information Processing Systems, 10220–10230.

[2] Lakkaraju, H.; Arsov, N.; and Bastani, O. 2020. Robust Black Box Explanations Under Distribution Shift. International Conference on Machine Learning (ICML) .

**Final Rebuttal Confidence:**

3

**Final Rebuttal Justification:**

While the authors have addressed several of my concerns, I believe that paper needs to be significantly updated to better clarify the motivation and practicality of the ablation study. Hence I am maintaining my original rating.

**Justification:**

While the paper attempts to understand the impact of different components of the XAI (perturbation based methods) pipeline, the main contributions and insights are not very clear in this version.  The paper needs significant updates before being accepted.  Hence, I recommend rejection.

---

> ### Author Rebuttal · Authors · 2024-10-25
>
> Thank you for the insightful review. Below you will find our response to the weaknesses you identified and how we plan to address them.
>
> ## Clarity of novelty, purpose, and contributions
> From your and the other the review it is clear to us that the key issue with our manuscript is clarity. The reason for the work is that we realized that the pixel attributions used by RISE can actually be applied to most perturbation-based image attribution methods. To investigate this we perform an ablation study to investigate which part of the RISE pipeline actually is most impactful. This study is the major contribution and novelty of the work. While some parts of the ablation study (such as using Gaussian blur instead of bilinear) are novel as well, they are not major contributions. Then one of the key findings is that pixel attribution (as one of the parts tested in the ablation study) is very beneficial. Unfortunately, the work does not clearly separate the study being performed from the results found from it.
>
> We will address this by rewriting the manuscript to emphasize the study and the reason for it and then specify that pixel attribution is only one of the parts that is being investigated, but that it is also very impactful and can be applied to other methods common in the field as well.
>
> ## Comparison to gradient-based methods
> The work is an ablations study for the pipeline used by most perturbation-based approaches. Gradient-based approaches do not fit into this pipeline and can therefore not be part of the ablation study. While comparing different methods from different approaches is good for benchmarking we chose not to compare in this work due to (a) focus on the ablation study and (b) it is an unfair comparison to compare the average performance of every possible perturbation-based pipeline to a specific optimized gradient-based method. Instead, this work could be used to inform the creation of a better perturbation-based method which in turn has to be compared against the SOTA methods of other approaches.
>
> We will address this by making the ablation study nature of our work more apparent so as to focus the work on the perturbation-based pipeline rather than trying to compare it to SOTA methods.
>
> ## Other metrics
> We did not have time and space in the manuscript for more metrics and insertion/deletion metrics are what the original RISE method was evaluated on which is why we use SRG (an insertion/deletion metric). Log-odds ratio is another insertion/deletion metric. Specifically, it occludes (deletes) the top-X percent most important pixels and compares how the performance changes. Since the original prediction doesn't change with the explanation method this part of the equation can be ignored for comparisons between methods. In this case, log-odds ratio is included in the MIF metric which is, in turn, included in the SRG metric we already use. The difference is that log-odds only evaluates for a specific top-X percent whereas MIF evaluates the top-Z percent for a variety of Z between 0% and 100%.
>
> We can unfortunately not address this issue.
>
> ## Contrast with referenced articles
> [1] introduces CXPlain a method for creating attribution maps that do not belong to either perturbation-based or gradient-based approaches. Instead, CXPlain trains a model to attribute influence on the model output to the input space. Essentially this can be seen as learning a segmentation model that segments the impactful areas of the input (though this is a simplified version). Like Grad-CAM, [1] belongs to another group of approaches than the perturbation-based methods being examined in this work and is therefore not useful for inclusion in the ablation study. However, [1] does use occlusion to create the loss used in training their explanation model which could be examined in a similar way (though this would be orders of magnitude more expensive as a model would need to be trained and evaluated for each method).
>
> [2] introduces the ROPE attribution method and evaluates it on tabular data. ROPE is an attribution method and could be used in that part of the image attribution pipeline. ROPE learns a surrogate model (similar to LIME and SHAP) but uses adversarial training. The results in the work are promising and logistic ROPE could be used as one of the options for the attribution method in this work. We will consider it for future work.
>
> ## Limited use of the validation dataset
> This work tests many possible combinations of different parts of the perturbation-based attribution pipeline. As such many different pipelines had to be tested (637 to be exact. excluding those used to test implementation and parts that did not make it into the manuscript). The most expensive of those experiments are those using the 8000 sample size which means we make 8000 model calls per image. To limit the time and energy consumption of the evaluation we chose to only use one image per class (resulting in 2% of the validation set) during the evaluation. This still required over 4 billion model calls in total. We also verified that this gave no statistically significant difference (p<0.05) in results compared to using the entire validation set for a few pipelines. Additionally, it is common to use even fewer images in the evaluation of XAI methods for images due to the computational expense.
>
> We will address this by more clearly motivating our choice of using only 1000 images for evaluation like described above.

---

### Meta-Review · Area_Chair_ERJW · 2024-11-02

**Recommendation:** Reject
**Confidence:** 4

**Metareview:**

The paper proposes an analysis of perturbation-based image explanations by testing multiple combinations of the different building blocks of the pipeline (segmentation, perturbation, etc.). Both the approach and most of the experimental design choices are sensible.

All the reviewers agree on the relevance of the topic and offered constructive advice on the approach and its evaluation. As a suggestion, it would be good to discuss the scope and limitations earlier on in the paper to help the reader understand and narrow down the different choices that come later with the experimental design.

However, even thought the authors managed to address many of the issues raised by the reviewers, there are some that remain after the rebuttal phase. The main one is around clarity (pointed by the reviewers but also acknowledged by the authors, and particularly important for the discussion in the paper). The manuscripts and its presentation could use further work and this is an aspect that would greatly improve the paper if properly tackled. The authors hinted at a few possible ways to change the manuscript, but it is not fully clear how a final version of the paper would look like — which makes the current assessment difficult.

Additionally, there are some other comments that were not addressed in full (e.g., lack of details around Table 2, use of synthetic data for evaluation) or not to the full satisfaction of the reviewers (size/choice of the dataset or statistical significance).

**Suggested Changes To The Recommendation:**

3: I agree that the recommendation could be moved up

---

### Decision · Program_Chairs · 2024-11-06

Reject